# ImmunoPET Directed to the Brain: A New Tool for Preclinical and Clinical Neuroscience

**DOI:** 10.3390/biom13010164

**Published:** 2023-01-13

**Authors:** Ángel García de Lucas, Urpo Lamminmäki, Francisco R. López-Picón

**Affiliations:** 1PET Preclinical Imaging Laboratory, Turku PET Centre, University of Turku, 20520 Turku, Finland; 2MediCity Research Laboratory, University of Turku, 20520 Turku, Finland; 3Department of Life Technologies, University of Turku, 20520 Turku, Finland

**Keywords:** positron emission tomography, neuroimaging, immunoPET, brain, blood–brain barrier, antibody engineering, neuroscience, neuro-oncology, neurology, neuropsychiatry

## Abstract

Immuno-positron emission tomography (immunoPET) is a non-invasive in vivo imaging method based on tracking and quantifying radiolabeled monoclonal antibodies (mAbs) and other related molecules, such as antibody fragments, nanobodies, or affibodies. However, the success of immunoPET in neuroimaging is limited because intact antibodies cannot penetrate the blood–brain barrier (BBB). In neuro-oncology, immunoPET has been successfully applied to brain tumors because of the compromised BBB. Different strategies, such as changes in antibody properties, use of physiological mechanisms in the BBB, or induced changes to BBB permeability, have been developed to deliver antibodies to the brain. These approaches have recently started to be applied in preclinical central nervous system PET studies. Therefore, immunoPET could be a new approach for developing more specific PET probes directed to different brain targets.

## 1. Introduction

Immuno-positron emission tomography (immunoPET) is a non-invasive in vivo imaging method based on tracking and quantifying radiolabeled monoclonal antibodies (mAbs) and other related molecules, such as antibody fragments, nanobodies, or affibodies [1,2]. Antibody imaging provides a specific and sensitive means of non-invasively characterizing the cell surface phenotype in vivo, which aids in diagnosis, prognosis, therapy selection, and monitoring of treatment for many diseases [3]. Since its inception, the field of immunoPET has been focused on oncology research because mAbs play an important role in the clinical management of cancer [4,5]. In addition to cancer, this technology is attractive for improving our knowledge in the diagnosis of brain disorders, as it is difficult to obtain biosamples from the brain [6]. However, the success of immunoPET in neuroimaging is limited because intact antibodies poorly penetrate the blood–brain barrier (BBB). Due to the poor permeability of the BBB, the delivery of therapeutic bioactive macromolecular compounds to the central nervous system (CNS) is difficult to achieve [7]. For example, only 0.1% of peripherally administered antibodies are delivered into the mouse brain [8]. To overcome this problem, different non-invasive methods have been developed to enhance antibody delivery to the CNS.

PET imaging studies of the CNS initially focused on global or regional changes in brain function, such as glucose utilization, cerebral blood flow, and oxygen metabolism. However, according to our knowledge of the diversity of neurotransmitter systems has been growing, we need to develop molecular imaging probes with an even higher target specificity than first-generation PET radiotracers [9]. Similar to CNS drug discovery, PET radiotracer development for the brain is scientifically challenging, with success being quite sporadic. Primarily, the radiotracer must show high affinity and selectivity for the target protein. Moreover, the radiotracer must be able to reach its target in vivo [10]. Therefore, the immunoPET approach could accelerate PET research in the brain to increase the development of new and more specific imaging probes.

The purpose of this literature review is to outline the major contributions of immunoPET to preclinical and clinical neurosciences. First, we write about strategies to cross the BBB, antibody design, and radiolabel. After that, we summarize the principal advances in neuro-oncology and neurological diseases. Finally, we hypothesize the development of immunoPET in these areas and the future possibilities for neuropsychiatry. Other extensive recent reviews can be found in the diagnosis of glioblastoma diagnosis [11] and Alzheimer’s disease research [6].

## 2. Strategies to Cross the BBB

Different strategies have been used to deliver antibodies into the brain, including modifications in their physicochemical properties; physiological mechanisms, such as adsorption-mediated transcytosis (AMT) or receptor-mediated transcytosis (RMT); and induced changes in BBB permeability (Table 1).

One physicochemical modification strategy is to increase the circulation half-life of antibodies, such as coupling poly(ethylene glycol) (PEG) to antibodies [12]. Prolonged circulation results in greater serum concentrations, which drive a greater accumulation of antibodies in low-permeability organs, such as the brain. In addition, PEG can act as a ligand for cholesterol transport receptors to promote the active transport of macromolecules into the brain [13].

Various cationic proteins can cross the BBB through AMT [14]. Upon electrostatic interactions between the cationized protein and anionic charges present on the surface of epithelial cells of BBB, the endocytosis of cationized proteins is triggered, a preliminary step for the protein to cross the BBB [15]. The variable heavy-chain domain of camel homodimeric antibodies (VHH) or nanobodies with a high isoelectric point (pI) have demonstrated an ability to cross the BBB through this mechanism [15,16].

RMT begins with the binding of a ligand to its cognate receptor on the luminal membrane of the brain‘s microvascular and capillary endothelial cells. It is a multistage process involving receptor-mediated endocytosis, mediated by clathrin-coated or non-clathrin-coated vesicles, followed by intracellular trafficking and vesicular sorting, finally resulting in vesicle fusion with the abluminal membrane of the BBB and delivery of the contents to the brain parenchyma [17]. Several receptors capable of inducing RMT are present in the BBB, such as the insulin receptor, transferrin receptor (TfR), and receptors responsible for lipoprotein transport, whereas others, such as the albumin receptor, are not expressed [18]. Antibodies can be modified to pass into the brain by conjugation with specific BBB receptor ligands [19]. For instance, antibodies targeting the transferrin receptor have been reported to cross the BBB [20]. Bivalent binding to TfR induces lysosomal sorting and degradation consistent with the incomplete transcellular trafficking observed in vivo, but monovalent binding leads to successful transcytosis in the BBB [21]. This brain shuttle strategy has been used successfully for immunoPET in animal models of Alzheimer’s, which are reviewed below [22,23,24,25,26,27,28,29,30,31,32]. 

The first attempt to modify the permeability of the BBB was in 1981 using mannitol, which was administered together with a drug, methotrexate, to enhance its delivery to brain tumors [33]. In the same way, other solvents, such as ethanol, dimethylsulfide, etoposide, or histamine, have been used to open the BBB and facilitate the delivery of drugs to the brain. However, the opening of the BBB is nonselective; thus, the use of these agents to affect the permeability of the BBB can cause serious side effects [19]. A more non-invasive, local, and targeted way to disrupt the BBB is with focused ultrasound (FUS). This technique, in combination with microbubbles, can lead to the transient and focal opening of the BBB, enabling the passage of therapeutic agents across the BBB without relying on the enhanced permeability and retention effect (EPR) [34]. Antibodies have successfully been delivered into the brain using FUS [35,36,37,38]. In addition, two studies have used an immunoPET approach to detect radiolabeled antibodies in gliomas [37,38].

## 3. Antibody Engineering Strategies for ImmunoPET 

Antibodies‘ capacity to cross the BBB, which is inherently limited for native antibodies [39], can be significantly improved by endowing the antibody with the capability to use RMT. This is achieved by generating bispecific antibodies that can bind both the RMT-associated receptor, such as TfR or insulin receptor and the target of interest in the CNS. The modular structure of the antibody molecule makes it possible to create bispecific antibodies of many different designs [40], including asymmetric IgG-like constructs in which the binding specificity of one of the two inherently identical binding sites has been altered [41,42,43]. In another common type of construct, antibody fragment(s) providing the second binding specificity are genetically fused to IgG [24,44,45]. Typically, single chain Fv (scFv) fragments consisting of the light and heavy chain variable domains linked together with a peptide linker of approximately 15–20 amino acids are used in such constructs. Yet another way to generate bispecific constructs is to conjugate two or more antibody fragments with different binding specificities together with peptide linkers [29,46,47]. Figure 1 shows an example of each of these three types of the bispecific antibody. Recently, Kariolis et al. reported a strategy to endow IgG with a capacity to bind TfR for RMT without modifying or adding variable domains for the second binding specificity. Instead, they engineered the surface of the *C*-terminal domain of the Fc-part to introduce a binding site for TfR [48]. In addition to the use of genetic engineering, chemical conjugation has also been employed for generating bispecific binders [22].

In addition to IgGs or fragments of them, other types of bioaffinity proteins can be utilized for the generation of bispecific targeting proteins, including single immunoglobulin domain-based binders obtained from camelid animals [49] or sharks [31] and alternative protein scaffolds, such as affibodies [2] and DARPins [50]. For bispecific binding, two or more such binders can be linked together with peptide linkers, or these binders can be linked to IgG instead of scFvs. In addition to being smaller (5–20 kDa) than scFvs, the alternative scaffold-based binders present various other benefits over antibody fragments, including higher stability and good expression properties. Overall, the ability to use a variety of antibody formats, alternative binders, and their potential combinations opens up a plethora of options for developing bispecific bioaffinity molecules with different types of structural and functional characteristics [51]. 

The bispecific designs can be assigned to two categories based on the presence or absence of the Fc-part of immunoglobulin. The presence of the Fc-part increases the circulation half-life of the antibody construct considerably due to FcRn-mediated rescue from lysosomal degradation. The Fc also contributes to the size of the construct by approximately 50 kDa. Overall, the Fc-part, whether present or absent, has a significant effect on the distribution and clearance of antibody constructs in the body [52] and affects the performance of antibody-based imaging agents.

Another important feature to consider in the bispecific designs is the binding valency (e.g., the number of binding sites for a specific antigen in the construct). Typically, there is either one or two binding sites of each specificity in a bispecific construct. Due to the bivalency of native IgG, the use of an asymmetric design is typically required to obtain monovalent binding with an IgG-based construct (Figure 1A). Compared to monovalent binding, bivalent binding can significantly increase the strength of the interaction with multimeric targets [53]. On the other hand, even in the presence of more than one binding site of certain specificity, the dimensions and flexibility of the constructs can affect the availability of the binding sites within it and, thus, the antibody´s mode of binding. For example, a construct with two TfR binding scFvs linked to the *C*-terminus of the light chains of IgG with short peptide linkers exhibits monovalent interaction with the dimeric TfR, suggesting that the limited flexibility of the construct hinders the simultaneous binding of the two scFvs [24].

Complex artificial antibody designs, typical in bispecific antibodies, tend to have an elevated risk of liabilities related to their biochemical or biophysical properties, including limited stability, low expression yields, a tendency to aggregate or poor pharmacokinetics [51,54]. These issues can sometimes be tackled or alleviated by further optimizing the antibody via genetic engineering, but the process can be time-consuming.

Genetic engineering can also be used to modify the binding sites of the antibody to optimize the strength of the binding. The two different binding sites of a bispecific antibody can differ in terms of the optimal binding affinity. When it comes to binding the actual imaging target, high-affinity binding is generally considered to be beneficial for successful immunoPET imaging. The situation can be different with the recognition of the RMT target; too high an affinity towards TfR can make the antibody stick to the receptor too tightly [20], increasing the likelihood of its lysosomal degradation instead of RMT. On the other hand, a too-weak affinity can impede the accumulation of antibodies in the cells of the BBB.

## 4. Radiolabeling Strategies

PET scans detect positron-emitting radionuclides, which are attached to a molecule such as a protein or an antibody. The positron is created, being short-lived, and eventually gets annihilated, converting all its mass into energy and thereby emitting two photons of 511 keV each (which is the resting energy of the electron or positron) in opposite directions. These two photons are detected in coincidence by scintillation detectors of PET scan [55]. 

Key to the labeling of antibodies, and antibodies-related molecules, is the appropriate matching between the biological half-life of the protein and the physical half-life of the radionuclide [2,56]. The shorter-lived PET radionuclides ^11^C, ^18^F, ^68^Ga, ^44^Sc, and ^64^Cu have been used for radiolabeling antibody fragments, while the slow pharmacokinetics of intact antibodies have enabled the use of the long-lived nuclides ^89^Zr and ^124^I [2]. PET radionuclides characteristics used to label antibodies and antibody fragments are summarized in Table 2.

The labeling of proteins can be performed by direct labeling, the addition of a prosthetic group, or using bifunctional chelators. Direct labeling, which nowadays involves mostly radioiodination, is the method used to label proteins without using intermediates such as prosthetic groups or bifunctional chelators. Prosthetic groups are small molecules able to bind with radionucleides in one site of the structure and simultaneously with a protein at a second site. An example of this procedure is the modification of the protein to bear unnatural biorthogonal functional groups such as an azide and then by using “click” chemistry to achieve the radiolabeled biomolecule. Radiometals (^68^Ga, ^44^Sc, ^64^Cu, and ^89^Zr) require bifunctional chelators, which are capable of coordinating a metal ion, and can be attached to the proteins. There are two main types of bifunctional chelators: cyclic, such as 1,4,7,10-tetraazacyclododecane-1,4,7,10-tetraacetic acid (DOTA) and 1,4,7-triazacyclononane-1,4,7-triacetic acid (NOTA); and acyclic multidentate, such as desferrioxamine B (DFO) and diamine-pentaacetic acid (DTPA) [59]. DOTA and NOTA chelation methods have been the mainstay for ^64^Cu labeling of antibody fragments. The DFO ligand is commonly used to chelate ^89^Zr due to its high chelation yields, mild reaction conditions, and reasonable stabilities [2]. In addition, several studies have shown that DFO, as well as NOTA, are suitable chelators for radiolabeling of biomolecules with ^68^Ga [2,60]. Finally, CHX-A″-DTPA (*N*-[(*R*)-2-amino-3-(para-isothiocyanato-phenyl) propyl]-trans-(*S,S*)-cyclohexane-1,2-diamine-*N,N,N′,N″,N″*-pentaacetic acid) has been identified as the most promising choice for ^44^Sc, as it permits labeling at room temperature within a reasonable period of time [61]. 

The long-lived radionuclides, such as ^89^Zr and ^124^I, are compatible with the long circulation of antibodies or antibody fragments, but they can result in high radiation burden and low image quality due to the long positron range [62]. To try to overcome this, a pre-targeting approach allows combining long circulating antibodies or antibody fragments with short-lived PET radionuclides such as ^18^F or ^11^C. In the pre-targeting approach, an antibody is chemically tagged, injected, and allowed enough time to accumulate in the target regions, and washout from non-target tissues and blood. Afterward, a radiolabeled small molecule that can rapidly react with the tag in the antibody is injected [63,64,65]. The combined use of the “click” reacting trans-cyclooctene (TCO) derivatives as chemical tags, and tetrazine (Tz) derivatives as radiolabeled small molecules, are the state of the art for pre-targeted PET imaging [66].

Recently developed bi-specific antibodies that penetrate the BBB by targeting transferrin [22,23,24,25,26,27,28,29,30,31,32] could be labeled with TCO for a pre-targeting approach across the BBB. The various ^18^F-labeled Tzs used for pre-targeting approaches in rodent cancer models and that could be used for neuroimaging studies, unfortunately, do not cross the BBB due to unfavorable characteristics [67]. Very recently, Shalgunov et al. developed a series of ^18^F-labelled Tzs that successfully penetrated the BBB in rodents and clicked in vivo with a TCO-polymer injected into the brain. The lead compound in this study is a promising tool for future pre-targeted neuroimaging studies [68].

## 5. Current Applications

### 5.1. Neuro-Oncology

In 2010, brain metastases were first visualized in HER2-positive breast cancer patients using [^89^Zr]Zr-DFO-trastuzumab. This study demonstrated brain lesions with an 18-fold higher [^89^Zr]Zr-DFO-trastuzumab uptake in tumors than in normal brain tissue. The brain penetration of [^89^Zr]Zr-DFO-trastuzumab was possible because of a disruption of the BBB at the site of the brain metastasis [69]. Another study in HER2-positive breast cancer patients using [^89^Zr]Zr-DFO-pertuzumab demonstrated the detection of brain metastases in these patients [70]. In HER2-negative metastatic breast cancer patients, immunoPET following a pre-targeting approach against carcinoembryonic antigen (CEA) showed higher overall sensitivity than [^18^F]FDG PET imaging in disclosing metastases, including brain dissemination [71]. In rodents, HER2-positive intracranial breast carcinoma xenografts have been shown to uptake of an ^18^F-labeled single-domain antibody fragment [72].

In addition to brain metastasis, most high-grade gliomas lead to the destruction of the BBB, with subsequent leakage of contrast medium, which is not shown by low-grade gliomas [73]. This increased permeabilization in high-grade gliomas, such as glioblastoma multiforme (GBM), could be used to deliver antibody-based therapeutics or immunoPET probes. In this sense, several preclinical studies have been conducted with different targets: integrin αvβ3 [74], epidermal growth factor receptor (EGFR) [61,75,76], delta-like ligand 4 (Dll4) [77], cluster of differentiation 105 (CD105) [75], fibroblast activation protein alpha (FAP) [78], CD146 [79,80], membrane-type 1 matrix metalloproteinase (MT1-MMP) [81], CD11b [82], vascular endothelial growth factor (VEGF) [83,84,85], transforming growth factor-β (TGF-β) [86,87], and prostate-specific membrane antigen (PSMA) [88,89]. In the early stages of this field, some studies with different immunoPET approaches were conducted in GBM flank xenograft models [61,74,75,77,78,79]. In addition to these models, orthotopic xenograft mice models were used to demonstrate the capacity of [^64^Cu]Cu-NOTA-YY146 to penetrate the disrupted BBB and efficaciously target CD146 within brain tumors, demonstrating that expression levels of CD146 can be scrutinized non-invasively in high-grade gliomas with PET imaging for potential patient selection and stratification for targeted therapies [80]. Another study that used orthotopic GBM xenograft models demonstrated the feasibility of using an immunoPET probe ([^89^Zr]Zr-DFO-LEM2/15) on MT1-MMP marker to visualize GBM tumors for diagnostic purposes. [^89^Zr]Zr-DFO-LEM2/15 was able to detect orthotopically growing GBM implants from TS543 (Figure 2A) but not U251, which correlates with the integrity of the BBB, as analyzed by Evans blue staining [81]. Therefore, the integrity of the BBB is one of the major limitations for the use of antibodies in high-grade gliomas because all GBM have clinically significant regions of tumor with an intact BBB [90]. In addition to targeting tumor cells, tumor-associated myeloid cells (TAMCs) can be an interesting immunoPET target because, in GBM, the immunosuppression is largely mediated by these cells. The [^89^Zr]Zr-DFO-anti-CD11b antibody PET imaging probe enabled the surveillance of immunosuppressive TAMCs in the tumor microenvironment of a mouse orthotopic glioma model, thereby providing a means of assessing the efficacy of immunotherapy [82]. 

One interesting example of fast translation from the preclinical to the clinical field is [^89^Zr]Zr-DFO-bevacizumab PET imaging in diffuse intrinsic pontine glioma (DIPG) [83,84,85]. Bevacizumab is a recombinant humanized monoclonal antibody against VEGF that has demonstrated significant responses and prolonged survival in individual patients with DIPG [91,92]. Therefore, the challenge is to identify patients who will benefit from treatment with bevacizumab [84]. In the first research paper, [^89^Zr]Zr-DFO-bevacizumab was used to study its biodistribution in different intracranial and subcutaneous murine tumor models, and accumulation of this antibody was only observed in the subcutaneous tumor models. These results are in line with the poor clinical response rates obtained with bevacizumab thus far in children with DIPG. Consequently, this research suggests that bevacizumab treatment is only justified if the tumor has been previously detected using [^89^Zr]Zr-DFO-bevacizumab [83]. One year later, a first pilot immunoPET study in pediatric DIPG patients suggested that the addition of [^89^Zr]Zr-DFO-bevacizumab PET imaging (Figure 2B) may help in selecting potential candidates for bevacizumab treatment of DIPG because this procedure assesses both target availability and drug accessibility of the tumor [84]. Subsequently, a study was performed with a 1-on-1 analysis of multiregional in vivo and ex vivo [^89^Zr]Zr-DFO-bevacizumab uptake, tumor histology, and vascular morphology in a DIPG patient. PET imaging was capable of detecting heterogeneity in [^89^Zr]Zr-DFO-bevacizumab uptake between lesions, which correlated well with the ex vivo measurements. However, PET cannot detect subcentimeter intralesional uptake heterogeneity [85]. Another clinical study demonstrated that a radiolabeled antibody against TGF-β, [^89^Zr]Zr-DFO-fresolimumab, reaches recurrent high-grade gliomas. However, monotherapy with fresolimumab did not result in an antitumor effect [86]. Interestingly, a recent preclinical research paper in which [^89^Zr]Zr-DFO-fresolimumab was used to detect radiation-induced TGF-β activation in tumors suggested that fresolimumab could improve the outcome of radiotherapy [87]. Finally, anti-PSMA radiolabeled antibody, [^89^Zr]Zr-DFO-huJ591, was studied in a 51-year-old woman diagnosed 8.5 years ago with grade II oligodendroglioma (1p/19q co-deleted, IDH mutant) and disease progression despite multiple prior treatments, including surgery and chemo-radiation, most recently with IDH inhibitor AG-120 (NCT02074839). [^89^Zr]Zr-DFO-huJ591 uptake was observed in two brain lesions determined to be anaplastic oligodendroglioma (grade III) histologically [88]. In addition to intact antibodies in clinical research, a radiolabeled minibody against PSMA, [^89^Zr]Zr-DFO-IAB2M, was tested in two high-grade gliomas and metastatic brain tumors from lung cancer patients. [^89^Zr]Zr-DFO-IAB2M may have potential value in the differential diagnosis of high-grade glioma from primary CNS lymphomas (PCNSL) or radiation necrosis, as well as in the prediction of treatment efficacy and assessment of the treatment response to bevacizumab therapy for high-grade glioma [89].

### 5.2. Neurological Diseases

The first attempts to reach the brain with a preclinical immunoPET approach used poly(ethylene glycol) antibodies against amyloid-β (Aβ). ^64^Cu-labeled anti-Aβ mAbs 6E10, M116, and M31, showed differences in uptake between the TgCRND8 mouse model of Alzheimer’s disease and wild-type animals [12,13]. In addition, radiolabeled antibodies without specific modification to cross the BBB were studied in Alzheimer‘s rodent models. However, limited penetration made them inadequate for monitoring cerebral amyloid pathogenesis [93,94]. 

The first successful bispecific antibody-based brain PET study was performed in 2016. A human version of mAb158 F(ab’)2 fragment against soluble Aβ protofibrils was chemically conjugated to 8D3 TfR antibody and radiolabeled with iodine-124. [^124^I]I-8D3-F(ab´)2-h158 was taken up into the brain through TfR-mediated transcytosis, and Aβ was detected in the tg-ArcSwe and tg-Swe Alzheimer mouse models [22]. Age or genotype did not influence the systemic pharmacokinetics or biodistribution to major organs [23]. In these studies, PET imaging correlated with age and Aβ pathology in the brains of Alzheimer‘s mouse models [22,23]. However, the chemical conjugation resulted in a heterogeneous mixture of fusion proteins randomly linked together at different positions, which is not suitable for clinical application. In addition, the use of the complete 8D3 antibody resulted in bivalent TfR binding, which has been shown to be suboptimal for transcytosis. Therefore, a new format was created in which the *C*-terminal end of the RmAb158 light chain was attached to the scFv of 8D3. In this way, monovalent interactions were achieved with TfR despite having two binding sites, improving transfer across the BBB, with higher or equal brain uptake compared to previously reported BBB shuttles at both trace and therapeutic doses [24]. [^124^I]I-RmAb158-scFv8D3 demonstrated an ability to follow disease progression and detect the effects of β-secretase inhibitor treatment by PET imaging in different mouse models of Alzheimer’s disease and was able to detect quantitative differences between treated and untreated groups, whereas [^11^C]Pittsburgh compound B ([^11^C]PiB) PET did not detect any differences between treated and untreated groups [25,26]. RmAb158-scFv8D3 and 8D3-F(ab’)2-h158 are large proteins (203 kDa and 270 kDa, respectively) with intact Fc domains and have long biological half-lives in blood, which make them not well suited for clinical use as radioligands [29]. For example, RmAb158-scFv8D3 was radiolabeled with fluorine-18, and PET imaging was performed 12 h after injection, but it was not enough to detect differences between transgenic mice with Aβ deposits and wild-type mice [27]. For this reason, smaller recombinant formats without the Fc domain, such as Tribody^TM^ (100 kDa) or di-scFv (58 kDa), have been created [28,29,30]. Tribody^TM^ radioligand is composed of two fragments of mAb158 fused to Fab 8D3. It has a shorter half-life in blood than RmAb158-scFv8D3 and 8D3-F(ab’)2-h158 (9 h vs. 12 h and 19 h, respectively) and shown a clear distinction between wild-type and Alzheimer mouse models [27,28]. [^124^I]I-Di-scFv3D6-8D3 was designed with an scFv derived from an anti-Aβ murine 3D6 antibody. It is capable of crossing the BBB and detecting in vivo intrabrain Aβ with better sensitivity than [^11^C]PiB [29]. [^125^I]I-Di-scFv3D6-8D3 has been measured at similar brain parenchymal concentrations as [^125^I]I-mAb3D6-scFv8D3, but with less retention in the capillaries, probably due to the lower avidity of [^125^I]I-di-scFv3D6-8D3 against TfR. The elimination rate from the brain is similar for both [30]. Interestingly, a single domain shark antibody VNAR fragment (TXB2) with similar affinity to murine and human TfR1 was studied in PET because 3D8 is an antibody against murine TfR. It was fused to the Aβ antibody bapineuzumab (Bapi) and showed markedly increased brain uptake compared to Bapi alone [31]. Furthermore, a recent study with variants of the rat TfR antibody, OX26, chemically conjugated to F(ab´)_2_ fragments of Bapi has demonstrated that the TfR-mediated transport of an immunoPET radioligand enables sensitive imaging of brain Aβ pathology in a rat model of AD (Figure 2C) [32].In addition to beta-amyloidopathy, another proteinopathy, α-synuclein (αSYN) deposition, has been studied by immunoPET. Several radiolabeled bispecific antibodies were positively studied in an αSYN deposition model with PET, and the in vitro characterization showed that the antibodies had high sensitivity towards aggregated αSYN. However, the lack of signal in the transgenic models (L61 and A30P) could be due to the intracellular localization of the target [95]. Finally, an apolipoprotein E (ApoE)-derived brain shuttle peptide fused to an ^18^F-labeled affibody against monomeric and oligomeric states of αSYN was successfully synthesized. The results of this study demonstrated that this strategy can potentially be utilized for brain molecular imaging [96].

ImmunoPET of neuroinflammation has been studied in the context of Alzheimer’s disease. A bispecific antibody based on the triggering receptor expressed on myeloid cells 2 (TREM2), a marker of microglial activation, and 8D3 scFv against transferrin was radiolabeled with iodine-124. Radiolabeled mAb1729-scFv8D3_CL_ showed differences in ex vivo autoradiography but not in PET. Therefore, other antibody formats or high-affinity antibodies must be developed to obtain clear PET imaging [97]. 

Multiple sclerosis (MS) is an immune-mediated neurological disorder in which immunoPET has been used. Anti-CD20 mAb has shown promising results in patients with relapsing-remitting MS [98,99,100]. However, no clinical tool exists to evaluate whether there is a preferential benefit in patients with B-cell disease. Recently, two preclinical research studies aimed to develop antibody-based PET probes against CD20 [101,102]. In early research, [^64^Cu]Cu-DOTA-rituximab showed human B cells in the spinal cord and brain of humanized autoimmune encephalomyelitis (EAE) mice model [101]. In the other study, ^89^Zr-labeled-anti-CD20 mAb was injected subcutaneously or intravenously in EAE and control mice. Regardless of the route of administration, biodistribution to all major organs was consistent in EAE and control mice. However, initial tracer uptake was significantly higher in the draining lymph nodes and parts of the CNS when administered subcutaneously compared to intravenously in EAE mice [102].

## 6. Perspectives

The imaging diagnostic tools currently employed in neuro-oncology, computed tomography (CT) and magnetic resonance imaging (MRI), provide excellent anatomical information on the localization of brain lesions but are frequently not able to differentiate tumor tissue from other concurrent processes, such as inflammation, edema, or bleeding, resulting in under- or over-estimation of the actual extension of the tumor [81]. In addition, following radiation and/or chemotherapy, neuro-oncologists often encounter treatment-related changes, such as pseudoprogression or necrosis [103]. Another important issue in neuro-oncology is that the majority of GBM have gross tumor burden with an intact BBB that extends beyond the disrupted BBB. Therefore, successful treatment is only possible if an effective drug is delivered with adequate exposure to the entire population of targeted cells [90]. The development of new antibody-based PET probes that can cross the BBB could improve the diagnosis of brain tumors and also be a great tool for managing immune and radiation therapy. 

Brain PET imaging allows a varied approach by assessing several physiopathological pathways, including neuroinflammation, neurotransmission, and protein aggregation, in neurological diseases [104]. Translocator protein-18 kDa (TSPO) is an 18 kDa outer mitochondrial membrane protein that is upregulated in neuroinflammation and a gold standard for PET imaging of activated microglia [105,106]. However, TSPO is expressed in reactive astrocytes, and many TSPO tracers have low sensitivity to the A147T variant. In addition, available TSPO ligands do not distinguish between pro- and anti-inflammatory microglia, which may be differentially expressed at the onset and progression of a neuroinflammatory condition [107,108]. Therefore, immunoPET could help develop new PET tracers targeting other neuroinflammatory biomarkers, such as a cannabinoid-2 receptor, purinergic receptors, or triggering receptors expressed on myeloid cells, among others [107]. Regarding neurotransmission, immunoPET could allow the development of specific PET tracers for targets that have proven to be difficult using small molecules. For example, the possibility of obtaining in vivo information about the integrity of the cholinergic transmission by PET imaging is of major interest in studying the pathophysiology of various neurodegenerative disorders, such as Alzheimer’s disease and Lewy body dementia, and has major relevance in cholinergic drug development [109]. Nevertheless, one of its key elements, muscarinic acetylcholine receptors (mAChRs), does not have a PET tracer with clinical applications, which demands mAChR tracers with improved imaging properties. In addition, M3 and M5 mAChR subtypes do not have any tracer in an early stage of development [110]. Finally, Parkinson’s disease, a protein aggregation disease, remains a challenge because clinical features of the disease overlap with other neurodegenerative conditions, and diagnostic tests or biomarkers still do not allow for a definitive diagnosis from the earliest stages. In addition, there is a need to better define Parkinson’s disease subtypes, which not only have different clinical presentations and prognoses but also differ in the underlying disease mechanisms, calling for personalized treatment approaches [111]. In this context, immunoPET could help create highly specific imaging probes against biomarkers, such as αSYN and TAR DNA-binding protein 43 (TDP-43), for which there are currently no other clinical PET probes available. It could improve early diagnosis and personalized medicine in neurodegenerative disorders. 

Another field in which immunoPET could bring new perspectives is neuropsychiatry. No specific marker of a psychiatric disorder has been established thus far [112], and the development of antibody-based PET probes could help find them. For example, postmortem investigations of GABA-A receptors in schizophrenia have shown that the α1 subunit increases and the α2 subunit decrease their expression, whereas the α5 subunit has inconsistent results. However, the density of GABA-A receptors in schizophrenia does not appear to be abnormal in radioligand studies, as postmortem studies have demonstrated. Some authors and we believe that these negative findings can be explained by the lack of imaging tracers with high affinity and specificity for different subunits [113]. Therefore, the high specificity of antibodies could be used to detect each of the 19 subunits of GABA-A receptors. In this way, new immunoPET tools could be developed to look for new biomarkers in neuropsychiatric disorders.

Recently developed PET technologies, such as multiplexed PET (mPET), long axial field of view (LAFOV) PET/CT scanners, and hybrid PET/MR imaging, together with immunoPET probes, could lead to novel knowledge in CNS research and contribute to drug development. The use of mPET, which can detect two radioisotopes at the same time [114], could allow the differentiation of, for example, specific GABA-A receptor configurations, such as α1β2 or α1β3 subunits, at once. This information could give a better picture of the GABA-A receptor subtype expression in vivo in psychiatric disorders. LAFOV PET scanners have an extended axial FOV (> 1 m), enabling whole-body dynamic PET imaging, and have several benefits, such as increased sensitivity, whole-body kinetic analysis, and lower radiotracer dosing [115,116]. These devices allow physiological or pathophysiological brain–heart–kidney–liver–gut interactions to be studied [115]. Finally, hybrid PET/MR allows temporal and spatial matching datasets displaying different physiological and metabolic information about a disease process. This multimodal technology could provide an efficient means for acquiring images in neuro-oncology, neurodegenerative disease, epilepsy, cerebrovascular disease, psychiatry, and neurology and may show added benefit beyond separate acquisition and interpretation of PET and MRI alone [117]. 

Current advances in antibody engineering have achieved successful application of immunoPET to the brain. This achievement opens the doors to improving diagnosis and treatment in neuro-oncology, neurology, and neuropsychiatry.

## Figures and Tables

**Figure 1 biomolecules-13-00164-f001:**
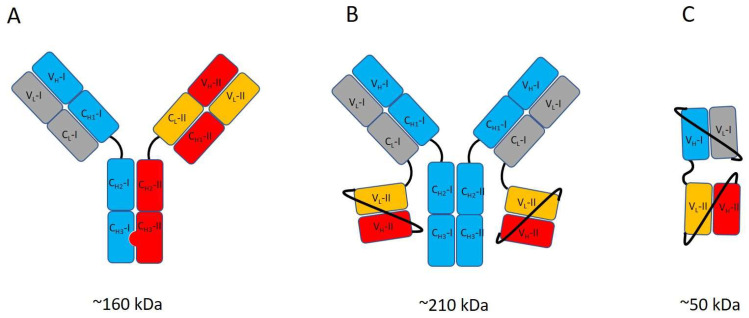
Examples of bispecific antibody constructs. (**A**) IgG-like asymmetric antibody engineered to bind an RMT-associated target with one of its binding arms and the CNS-associated target with the other binding arm. The knob-into-hole approach [41] is used to guide correct pairing of two different heavy chains, and CrossMab technology [42] is used to prevent mispairing of the light chains. (**B**) IgG-like antibody with scFv fragments of different binding specificity conjugated to the C-terminus of the light chains. This is a symmetric IgG-based construct. (**C**) di-scFv construct in which two scFvs with different binding specificities are fused together via a short peptide linker. V_L_ and V_C_ refer to the light chain variable and constant domains, respectively. V_H_, C_H1_, C_H2_ and C_H3_ refer to the heavy chain variable domain and the constant domains 1, 2 and 3, respectively. The roman numbers in the domain names (e.g., V_L_-I or V_L_-II) show which of the two different binding specificities the domain is associated to.

**Figure 2 biomolecules-13-00164-f002:**
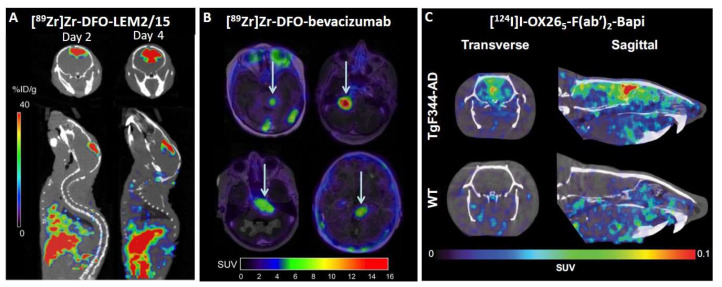
Examples of immunoPET images in preclinical and clinical neuroscience. (**A**) Representative fused PET/CT images of coronal and sagittal planes at 2 and 4 days post-injection containing TS543 brain tumors with [^89^Zr]Zr-DFO-LEM2/15. (Image: Adapted from Ref. [81]). (**B**) PET/MR fusion images of four different pediatric patients with DIPG where four tumors showed variable uptake of [^89^Zr]Zr-DFO-bevacizumab (arrows). (Image: Adapted from Ref. [84]). (**C**) Representative in vivo PET images from TgF344-AD and WT rats 3 days post-administration of [124I]I-OX26_5_-F(ab´)_2_-Bapi. (Image: Adapted from Ref. [32]).

**Table 1 biomolecules-13-00164-t001:** Summary of different strategies to cross the BBB with proteins or antibodies.

Strategy	Mechanism	References
Physicochemical properties modification	Poly(ethylene glycol) conjugation to increase the circulation half-life	[12,13]
Through physiological mechanism	Cationic proteins or nanobodies that trigger adsorption-mediated transcytosis (AMT)	[14,15,16]
	Ligands or antibodies that trigger receptor-mediated transcytosis (RMT)	[17,18,19,20,21,22,23,24,25,26,27,28,29,30,31,32]
BBB permeability changes	Open BBB with solvents such as mannitol	[19,33]
	Focused ultrasound (FUS) with microbubbles	[34,35,36,37,38]

**Table 2 biomolecules-13-00164-t002:** Summary of PET radionuclides characteristics used in immunoPET. Data extracted from [57,58].

Radionuclide	Half-Life	Branching Ratio (Β^+^) (%)	Positron Energy–E _Max_ [Mev]	Mean Positron Range (mm)
^11^C	20.4 min	99	0.97	1.2
^18^F	109.7 min	97	0.65	0.6
^68^Ga	67.7 min	89	1.90	3.5
^44^Sc	3.97 h	94	1.47	2.3
^64^Cu	12.7 h	18	0.65	0.7
^89^Zr	78.4 h	23	0.91	1.3
^124^I	100.2 h	23	1.54	4.4

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
