# Peer review of "ImmunoPET Directed to the Brain: A New Tool for Preclinical and Clinical Neuroscience"

_biomolecules, 2023, doi:10.3390/biom13010164_

Round 1
Reviewer 1 Report
Review Report
The authors have written a sound review paper on the potential application of ImmunoPET imaging of brain through tracking and quantification of radiolabeled monoclonal antibodies (mAbs) and other 10 related molecules, such as antibody fragments, nanobodies, or affibodies. Approaches for modifying the permeability of BBB have been stated comprehensively. This is addition to the descriptions of specific applications of ImmunoPET in managing brain related issues. However, I would like to suggest the following minor corrections before the acceptance of the manuscript for publication in the Biomolecules Journal.
1. The introduction part is very short and needs to be improved by adding a short summary related to the comparison between ImmunoPET and other imaging techniques. Additionally, a paragraph needs to be added at end of the introduction section describing the aims of the review paper and description of its composition.
2. In section 2, please prepare a comparison table to simplify the comprehension of the brain delivery strategies.
3. Apply comment 2 to section 3 as well.
4. Please consider moving statement in lines 155-157 to the introduction part.
5. In section 4, please include eye catching illustration to aid the better understanding of the reviewed applications.
6. Please merge sections 5 and 6
Author Response
Reviewer 1
1) The introduction part is very short and needs to be improved by adding a short summary related to the comparison between ImmunoPET and other imaging techniques. Additionally, a paragraph needs to be added at end of the introduction section describing the aims of the review paper and description of its composition.
Following the suggestion of the reviewer, we extended the introduction section adding two new paragraph, pags. 1-2 lines 39-55.
2) In section 2, please prepare a comparison table to simplify the comprehension of the brain delivery strategies.
Following the suggestion by the reviewer, we have introduced a table (table 1) in this section. Pag. 2 lines 61
3) Apply comment 2 to section 3 as well.
Due the very extensive variety of antibody formats, and given that an extensive explanation of them, is not the scope of this review, we have added the following paragraph with a reference to a dedicated review article to the topic.
“Overall, the ability to use a variety of antibody formats, alternative binders, and their potential combinations opens up a plethora of options for developing bispecific bioaffinity molecules with different types structural and functional characteristics [52]”. pag. 4 , lines 140-143
4) Please consider moving statement in lines 155-157 to the introduction part.
As suggested, we have moved the lines 155-157 to the introduction paragraph pag. 2 lines 54-55.
5) In section 4, please include eye catching illustration to aid the better understanding of the reviewed applications.
Following the suggestion, we have created an eye-catching illustration including images of different ImmunoPET studies carried out in humans and in disease models . Pag. 9 lines 378-392
6) Please merge sections 5 and 6
As suggested, we have now merged both sections.

Reviewer 2 Report
Please refer to the attached file, thanks!

Author Response
Reviewer 2
1) The authors should briefly introduce the strategies and considerations of radiolabeling and positron emitter selection in immunoPET related to brain imaging. In this way, the readers may better appreciate the motivation to use long half life radionuclides and the motivation to develop low molecular weight tracers.
A new section has been created for radiolabeling strategies together with a table (table 2). Pags. 5-6 lines 179-229.
2) In section 2, please briefly discuss the situations when the continuity of BBB is compromised, such as in cases of brain tumors. This corresponds to the description on Page 4 Lines 173–198. Alternatively, you may rename the section to “Strategies to cross the BBB”.
The tittle of the section has been modified as suggested.
3) Page 1 Line 41, cite the sources for AMT and RMT.
This paragraph was conceived as introduction to the text following where the specific references can be found.
4) The concrete concept of AMT is also not given. Please explicitly present the meaning of this concept.
The paragraph was probably confusing. For clarification, we re-wrote the text. pag. 2 line 71-72.
5) On Page 2 Line 66, it is stated that “In addition to these ligands”, however, TfR is included in the previous sentence. Please solve the contradiction. And please pay attention to the acronym of TfR, which should appear in the first time it was mentioned.
The text paragraph was slightly modified to solve the contradiction. pag. 3 line 82-86
6) Page 2 Line 90 to Page 3 Line 97, cite sources for the three different constructs of the bispecific IgG-related tracers, by the end of each sentence.
Following the suggestion, references [24, 42-48] have been added.
7) Page 3 Line 132, please cite the source for the multivalence effect.
References [54] has been added.
8) Page 4 Lines 141–145, please cite sources.
References [52,55] has been added.
9) Page 4 Lines 141–150, is this your opinion or a common knowledge? If it is your opinion, please specify (“It is generally believed that most CNS-related targets require affinity as high as possible”); otherwise, please cite sources to support your claim.
We re-wrote the text to clarify the concept. pag. 2 line 71-72.
10) Page 7 Line 299. I would not call MS a neurological disorder. Rather, it is more appropriate to be seen as an immune disorder affecting the nervous system.
We modified the text defining MS as an “immune-mediated neurological disorder” instead of neurological disorder. Pag. 10 lines 400-401.
